# pH-Sensitive Ratiometric Fluorescent Probe for Evaluation of Tumor Treatments

**DOI:** 10.3390/ma12101632

**Published:** 2019-05-18

**Authors:** Peisen Zhang, Junli Meng, Yingying Li, Zihua Wang, Yi Hou

**Affiliations:** 1Key Laboratory of Colloid, Interface and Chemical Thermodynamics, Institute of Chemistry, Chinese Academy of Sciences, Beijing 100190, China; zhangps@iccas.ac.cn (P.Z.); mengjl@iccas.ac.cn (J.M.); liyingying@iccas.ac.cn (Y.L.); 2School of Chemistry and Chemical Engineering, University of Chinese Academy of Sciences, Beijing 100049, China; 3College of Life Science and Technology, Beijing University of Chemical Technology, Beijing 100029, China

**Keywords:** ratiometric fluorescence, intracellular pH, pharmacodynamic evaluation

## Abstract

Determining therapeutic efficacy is critical for tumor precision theranostics. In order to monitor the efficacy of anti-cancer drugs (e.g., Paclitaxel), a pH-sensitive ratiometric fluorescent imaging probe was constructed. The pH-sensitive ratiometric fluorescent dye ANNA was covalently coupled to the N-terminal of the cell-penetrating TAT peptide through an amidation reaction (TAT-ANNA). The in vitro cellular experiments determined that the TAT-ANNA probe could penetrate the cell membrane and image the intracellular pH in real time. The in vivo experiments were then carried out, and the ratiometric pH response to the state of the tumor was recorded immediately after medication. The TAT-ANNA probe was successfully used to monitor the pharmacodynamics of anti-cancer drugs in vivo.

## 1. Introduction

Due to high risk, tumor theranostics has become the main focus and challenge in clinical medicine and medical investigations. Effective tumor therapy vastly depends on precision tumor diagnostics [1,2]. Over the past decades, medical imaging techniques have been developed and have become powerful tools for tumor detection, such as MRI, CT, and even optical imaging. However, in a clinical scenario, it is not only necessary to localize the position of the tumor, but also to evaluate tumor progression, therapeutic efficacy, etc. [3,4]. 

It is well known that tumor prognosis and therapeutic administration are strongly correlated with the extracellular microenvironment of the tumor [5]. For instance, due to anaerobic glycolysis, tumor tissues always have a weak acidic extracellular environment, which could facilitate the invasion, mutation, metastasis, and multi-drug resistance of the tumor [6,7]. On the other hand, some treatments can also alter factors of the microenvironment, such as the aggravation of acidity in residual tumor tissues after photothermal therapy, which might increase the resistance of surviving tumor cells [8,9]. However, extracellular pH changes are always downstream of cellular physiological or pathological processes, i.e., once this change can be discovered, the metabolic process of cells has already been completed. Therefore, determining how to predict changes in tumors earlier, especially in consideration of the fast evaluation of drug efficacy, is a prerequisite for directing therapeutic administration.

In contrast to extracellular pH, intracellular pH is a relatively stable physiological indicator of cells in different cellular processes [10,11,12]. High intracellular pH is required for cell proliferation and differentiation, especially in tumor cells. More importantly, intracellular pH tends to change in the early stages of metabolic processes, which can be used to predict some physiological or pathological behaviors. For instance, intracellular acidification has been reported to be an early feature of apoptosis in neutrophils [13,14]. This acidification is a prerequisite for DNA cleavage and possibly for other biochemical changes of apoptosis. Thus, knowledge of the change in intracellular pH can directly help us to determine therapeutic efficacy, predict metastasis potential, and direct personalized treatments.

At present, analysis of blood samples and occasionally biopsy specimens of relevant tissues are the "gold standard" for pharmacodynamic evaluation. However, blood tests with a limited index possibly cause false positive and false negative results, which may not truly reflect the growth state of cancer. In addition, a biopsy will destroy the tumor and surrounding normal tissue and increase the metastatic rate. Accordingly, it is necessary to invent a non-invasive and reliable way to evaluate the efficacy of anti-cancer drugs. Optical imaging techniques provide a facile and hypersensitive tool to detect early, tiny tumors in vivo [15]. Combined with probes that respond to physiological indicators of cells, it could be possible to thoroughly analyze these tumor-associated prognostic factors in real time. However, the fluorescence intensity, based on conventional fluorescent probes, always suffers from several interferences, such as self-quenching at high concentrations, autofluorescence from biological tissues, and different settings of instrumental parameters, which lead to deviations in the imaging results [16,17]. To avoid these interferences, ratiometric fluorescent probes have been developed [18,19]. Through the specific value, instead of absolute intensity, the probe is able to determine the factors in the tumor microenvironment. In previous works, we successfully mapped extracellular pH and MMP-9 activity of tumors in vivo, and further disclosed their relationships with tumor prognoses [20].

Herein, we report the findings of a novel ratiometric fluorescent pH probe based on a pH-sensitive ratiometric fluorescent dye, ANNA, and a cell-penetrating peptide, TAT (TAT-ANNA), to sensitively monitor the intracellular pH, thereby evaluating the pharmacodynamics of anti-cancer drugs in vivo. The chromophores are covalently coupled to the N-terminal of the TAT peptide through an amidation reaction. Through a series of in vitro and in vivo experiments, the performance of TAT-ANNA was evaluated. This probe successfully penetrated the cell membrane and imaged the intracellular pH. In addition, it could respond to the state of the tumor immediately after medication to provide a reliable pharmacodynamic evaluation. 

## 2. Materials and Methods 

### 2.1. Materials

4-(4,6-dimethoxy-1,3,5-triazin-2-yl)-4-methylmorpholinium chloride (DMTMM) and MTT(M2128) were purchased from Sigma-Aldrich. Human colorectal cancer cell lines LS180 were obtained from the School of Oncology at Peking University. The sodium pentobarbital was purchased from Baxter Healthcare Corporation. The TAT peptide was synthesized by the China Peptides Company. Paclitaxel Injection (PTX) and Cetuximab were purchased from Pfizer Incorporated Company.

### 2.2. Synthesis of TAT-ANNA (Scheme S1)

ANNA was synthesized according to our previously reported method [20,21,22]. ANNA (0.39 mg, 0.001 mmol) was dissolved in 200 μL of H_2_O, and DMTMM (0.348 mg, 0.0012 mmol) was added to activated carboxyl group with ANNA at room temperature for 10 min. Then, the TAT peptide (1.559 mg, 0.001 mmol) was added to the mixture and vibrated for 1 h to conjugate the TAT peptide to ANNA. The product was purified by using a 1K MWCO centrifugal device to remove the free ANNA.

### 2.3. UV–Vis Absorption and Fluorescence Spectra Measurement

TAT-ANNA and TAT peptides were dissolved in water and the absorption spectra were recorded from 200 nm to 800 nm. For the fluorescence spectra measurement, TAT-ANNA was dissolved in a series of different pH buffer solutions and the fluorescence spectra were measured from 465 nm to 650 nm under 455 nm excitation. 

### 2.4. Cell Culture and Specific Binding Assays In Vitro

The human colorectal cancer cell line LS180 was cultured in a medium of high-glucose DMEM and F-12K nutrient mixture (1:1), supplemented with 10% fetal bovine serum, 100 U mL^−1^ penicillin, and 0.1 mg mL^−1^ streptomycin at 37 °C under a 5% CO_2_ atmosphere.

Approximately 105 LS180 cells were seeded in the wells of confocal capsules and incubated overnight at 37 °C under 5% CO_2_ to allow a firm adherence. After being rinsed with PBS buffer, the cells were incubated with the TAT-ANNA probe and the ANNA dye with the same concentration (5.0 μM) in PBS solution, respectively, for 2 h at 37 °C. After that, the cells were rinsed three times with PBS and imaged by confocal microscopy (Olympus FV 1200, Tokyo, Japan), the fluorescence signals were collected at 500–545 nm under excitation at 458 nm.

### 2.5. Cytotoxicity of the TAT-ANNA Probes

MTT assays on LS180 cells were carried out as follows. Cells were seeded into a 96-well cell culture plate at 5 × 10^3^ cells/well under 100% humidity, and then cultured at 37 °C in an atmosphere containing 5% CO_2_ for 24 h. The TAT-ANNA probes were added to the wells at a range of concentrations, and incubated with the cells for 24 h at 37 °C under 5% CO_2_. Subsequently, the supernatant containing the excrescent nanoparticles was decanted, and the cells were incubated for another 48 h. After that, 10 μL of MTT (5 mg/mL) was added to each well, and incubated for 4 h at 37 °C. Thereafter, 150 μL of DMSO was added into each well, and the assay plate was shaken for 20 min. The optical density of each well at 490 nm was recorded on a microplate reader (Thermo, Varioskan Flash), while the optical density at 630 nm was used as a reference.

### 2.6. Intracellular pH Calibration

Approximately 10^5^ LS180 cells were seeded in the wells of confocal capsules and incubated overnight at 37 °C under 5% CO_2_ to allow a firm adherence. Then the cells were incubated with the TAT-ANNA probe PBS solution (5.0 μM) for 2 h. After that, the cells were rinsed three times with PBS and fixed in 4% paraformaldehyde for 30 min. The buffer solutions at a series of different pH levels were added in different confocal capsules to adjust the pH of fixed cells overnight, and to make sure that the intracellular pH was consistent with the surrounding buffer solution. The imaging of cells was carried out on a confocal microscope (Olympus FV 1200). The fluorescence signals of the cells were detected in two channels: the blue channel (465–500 nm) and the green channel (510–600 nm). 

### 2.7. Effect of PTX and Cetuximab on the Intracellular pH

LS180 cells were incubated with PTX (10 mg/L) and Cetuximab (20 mg/L) in DMEM at 37 °C for 2 h, respectively, and further incubated with 5.0 μM TAT-ANNA in PBS solution for 2 h. Then the TAT-ANNA stained cells were washed with PBS twice, and the fluorescence images were recorded immediately by a confocal microscope (Olympus FV 1200). 

### 2.8. Animal Tumor Models

The tumor models used were established upon subcutaneous injections of LS180 cells (∼5 × 10^6^) into 4 week old male BALB/c nude mice. The tumor imaging studies were carried out7 days after the inoculation of tumor cells.

### 2.9. Pharmacodynamic Evaluation of Anti-Tumor Drugs In Vivo

First, 100 μL TAT-ANNA probe and normal saline (NS) solution with the concentration of 4 mM were injected intratumorally. Then the tumor of the test mouse was treated with 100 μL PTX (NS solution, 1 mg/mL) (intratumoral injection), while the tumor in the control mouse was injected with NS solution of the same volume. The fluorescent images of the tumors were captured at different time points after the mice were anesthetized with 2% sodium pentobarbital intraperitoneal injections, using a digital camera equipped with 450–490 nm and 510–550 nm band-pass filters. The excitation light (365 nm) can cover a whole mouse with a density of 12 mW/cm^2^. 

### 2.10. Histological Study

After capturing the images, the mice were executed. The subcutaneous tumors were harvested and kept in 10% formalin for 3 days. After being embedded into paraffin, the fixed tumors were sliced and stained with hematoxylin and eosin (H&E), Ki 67, PCNA, Caspase-3, and then subjected to a microscopy study.

## 3. Results and Discussion

### 3.1. Construction of the pH-Sensitive Ratiometric Fluorescent TAT-ANNA Probe 

The design and synthesis of TAT-ANNA is shown in Figure 1a and Appendix A. TAT-ANNA is composed of a cell-penetrating peptide TAT (YGRKKRRQRRR) and a pH-sensitive naphthalimide dye (ANNA, the fluorescence spectra is shown in Appendix A). To construct the probe, TAT was used as a transduction domain which was covalently coupled with ANNA through the terminal amino group of each TAT and the carboxyl group of ANNA via amidation reactions. The absorption spectroscopy results, as shown in Figure 1b, reveal that after the conjugation reaction with ANNA, an absorption band appeared from 400 to 500 nm. This is the typical absorption of ANNA, which indicates that the construction of the probe was successful. Due to the deprotonation and internal charge transfer from the electron-rich amino group to the electron-poor imide moiety (Figure 1c), the probe presented a strong pH-dependent ratiometric emission, as shown in Figure 1d. The resulting intensity ratio of the fluorescence acquired at 480 and 510 nm (i.e., I_480_/I_510_) exhibited a sharp decrease, particularly in the range of 5.0–7.0 (Figure 1e). This is especially interesting for detecting intracellular pH.

### 3.2. Specific Binding Assays and Cytotoxicity In Vitro

TAT, a kind of cell-penetrating peptide, is derived from the human HIV virus TAT protein, which can result in the delivery of macromolecules into the cytoplasm. To validate the ability of TAT-ANNA probes to penetrate cell membranes, LS180 cells of human colorectal cancer cell lines were used for in vitro cellular experiments. As displayed in Figure 2a, the cells co-incubated with the TAT-ANNA probes show intense fluorescence signals. Notably, the overlay of bright-field images and the fluorescent images of cells reveal that the TAT-ANNA probes predominantly entered the cell and distributed throughout the cytoplasm. In contrast, the signal presented by the cells incubated with the unmodified ANNA dye were only located outside the cell and enclosed in the cell membrane. These results demonstrate that the TAT domain was successful and that the TAT-ANNA probe penetrated the cell membrane and entered into the cell. 

To confirm the cytotoxicity of the TAT-ANNA probe, cytotoxicity was investigated through methyl thiazolyl tetrazolium (MTT) assays on the proliferation of LS180 cells. The concentration of probes was determined by the UV–vis method. As shown in Figure 2b, no significant cytotoxicity was observed in the TAT-ANNA probes. More than 85% of the cells were still alive with a probe concentration of 8 mM, which is two times higher than the injection dose in mice (100 μL 4 mM per mice). This result indicates that the probe has no obvious side effects as a tumor imaging agent.

### 3.3. pH-Sensitive Ratiometric Fluorescent Imaging Based on TAT-ANNA In Vitro

The ratiometric pH imaging of TAT-ANNA was studied using confocal microscopy in two channels ascribed to the emission of ANNA in different wavebands. To determine the pH in cells, the valid intracellular pH calibration was carried out using buffer solutions at different pH values in fixed LS180 cells. As shown in Figure 2c, the fluorescence intensity in the blue channel (480 nm waveband emission of ANNA) significantly decreased against the increase of pH from 4.84 to 8.23, whereas the fluorescence intensity in the green channel (510 nm waveband emission of ANNA) showed the opposite trend, as it rose gradually along with the pH increase. The merged images showed obvious changes in color from blue to green as the pH increased. Moreover, the pseudo-color ratio images (ratio I480/I510), which were obtained from the 480 nm and 510 nm green channel fluorescent images above, show more obvious color changes with the variation of the pH.

In the process of cell death through various pathways, intracellular pH usually changes due to cell metabolism disorder or lysosomal membrane damage. Therefore, measuring intracellular pH changes can be an effective way to evaluate the efficacy of anti-cancer drugs. Two clinically used anti-cancer drugs, Paclitaxel Injection (PTX) and Cetuximab, were chosen to investigate the ability of the probes to evaluate drug efficacy (Figure 3). As a result, both of the anti-cancer drugs cause the remarkable decrease in intracellular pH after co-incubating at a low concentration, which is the prerequisite for apoptosis. This indicates that monitoring the intracellular pH is a decent way to evaluate whether anti-cancer drugs work on cancer cells.

### 3.4. Pharmacodynamic Evaluation In Vivo

With respect to pharmacodynamic evaluation in vivo, the subcutaneous tumor model was adopted to better show the optical response of the probe. The TAT-ANNA probe was injected into the tumor-bearing nude mice at the flank region of the right hind leg. This was conducted as a proof of principle to demonstrate the feasibility of applications with the probe. Color-coded fluorescence images of the tumors based on the emission of 450–490 nm bands (blue channel) and 510–550 nm bands (green channel) are shown in Figure 4a. After injecting the anti-cancer drug, PTX, the blue channel of the fluorescence signal increased significantly over a period of 6 h following the injection, while the green channel in the same area of tumor descended correspondingly. In contrast, the fluorescence intensity of the control mouse, which was only injected with normal saline (NS) without any anti-cancer drugs, remained constant in both the blue and green channels.

The fluorescent intensity changes of the two emission bands allowed us to map the pH of the tumor region so as to evaluate the efficacy of the anti-cancer drugs. The correct I_480_/ I_510_ ratio in the tumor site together with its pH dependency are shown in Figure 5a. After the injection of PTX, the acidity of the tumors increased gradually, which further revealed that the tumor cells had been affected by PTX, i.e., the anti-cancer drug had begun to take effect. The results above indicate that TAT-ANNA can be used as an excellent tool for anti-cancer drug evaluation.

In clinical trials, biopsies are the most accurate way to verify and observe the efficacy of drugs to provide reference for clinical medication. Therefore, histopathological and immunohistochemical studies were used to further confirm the reliability of TAT-ANNA probes in drug evaluation. Mice were sacrificed after in vivo pharmacodynamic evaluation and then the tumor tissues were harvested for H&E staining and immunohistochemical assays. The expressions of cell proliferation-related antigens Ki 67 and PCNA were significantly downregulated in the tumor tissues that received the anti-cancer drug PTX, contrasting to those that received NS (Figure 5b). Such results support the findings that PTX begins to restrict the growth of cancer cells, which was perfectly consistent with the evaluation of the probe. In addition, the immunohistochemical staining for Caspase-3 indicated that a great percentage of cells in the PTX-treated tumor suffered from apoptosis, which revealed the immediate cause for the decrease. All these histopathological results support the findings that the probe could provide a reliable evaluation of drug efficacy immediately after medication. Thus, the present design of the responsive ratiometric fluorescent probe provides an approach to non-invasively monitor treatment efficiency in real time, which should facilitate the theranostics of cancer.

## 4. Conclusions

In summary, by coupling a pH-sensitive fluorescent dye, ANNA, and a cell-penetrating peptide, TAT, a novel ratiometric fluorescent pH probe (TAT-ANNA) was developed. This probe can successfully penetrate the cell membrane and image the intracellular pH. More importantly, the sensitive response to pH variations provides great potential for the TAT-ANNA probe to be used as a drug evaluation tool. In vivo experiments showed that the probe could respond to the state of the tumor immediately after medication. The drug efficacy evaluated by TAT-ANNA was consistent with the results of the biopsy, which further displays the reliability of the probe in evaluating drug efficacy. In short, the current study reports a new type of ratiometric fluorescent pH probe, TAT-ANNA, that is potentially useful for sensitively imaging the intracellular pH, thus offering a novel way to achieve non-invasive pharmacodynamic evaluation.

## Figures and Tables

**Figure 1 materials-12-01632-f001:**
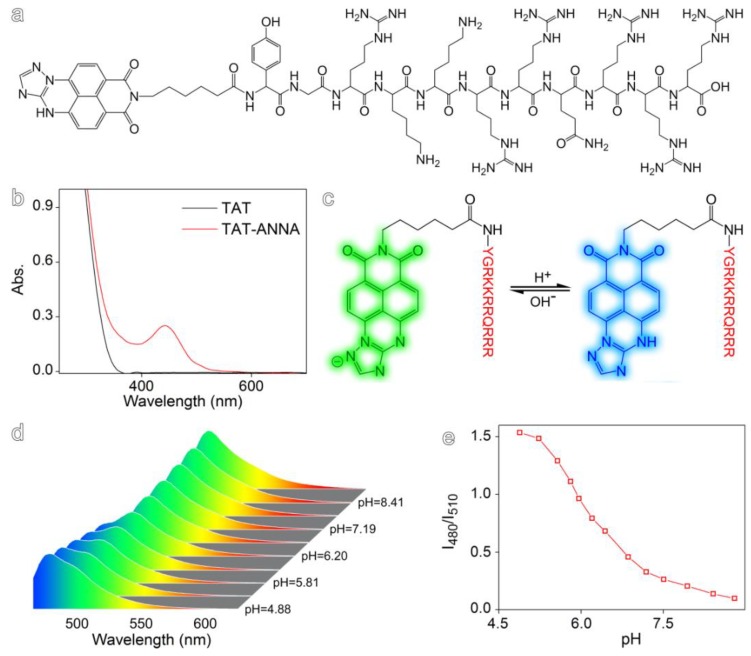
(**a**) Illustration of the TAT-ANNA probe. (**b**) Ultraviolet−visible absorption spectrum of the TAT peptide and the TAT-ANNA probe. (**c**) Protonation/deprotonation-induced structural transformation of the TAT-ANNA probe. (**d**) Fluorescence spectra of TAT-ANNA recorded at different pH values excited at 455 nm.

**Figure 2 materials-12-01632-f002:**
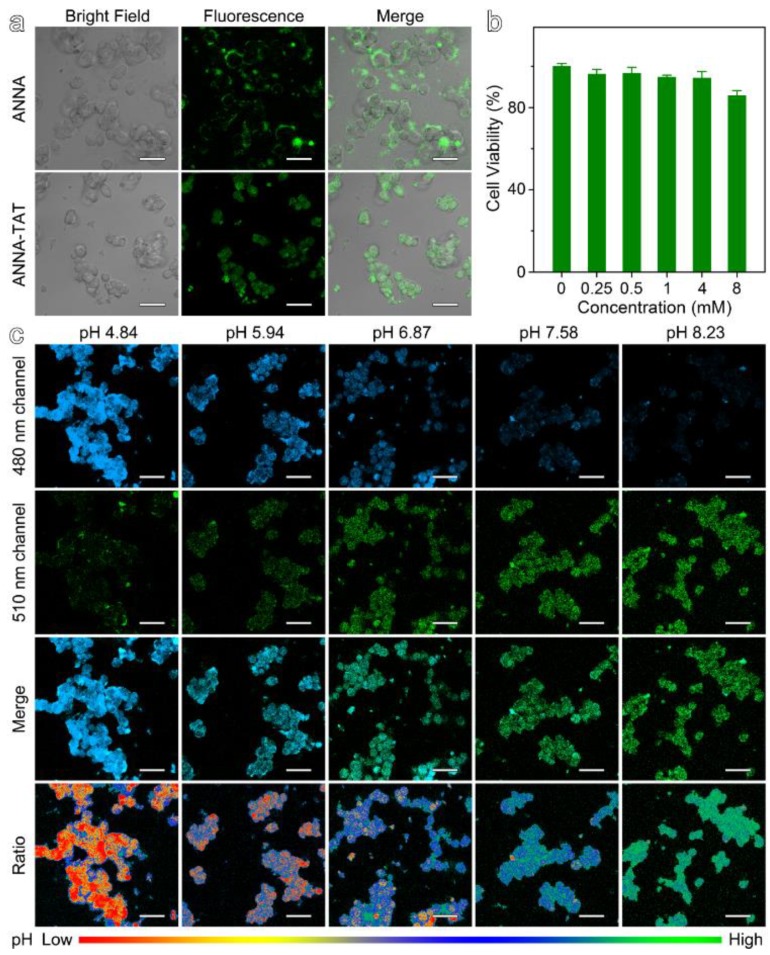
(**a**) Bright-field images and fluorescence images, together with merged images of LS180 cells incubated with TAT-ANNA and ANNA dye, respectively. (**b**) Viabilities of LS180 cells treated with TAT-ANNA probe. (**c**) Fluorescence images (collected at 465–490 nm and 510–550 nm for two channels), merged images, and ratio images of LS180 cells incubated with TAT-ANNA in buffer solutions with different pH values. The scale bars of (**a**) and (**c**) correspond to 10 μm.

**Figure 3 materials-12-01632-f003:**
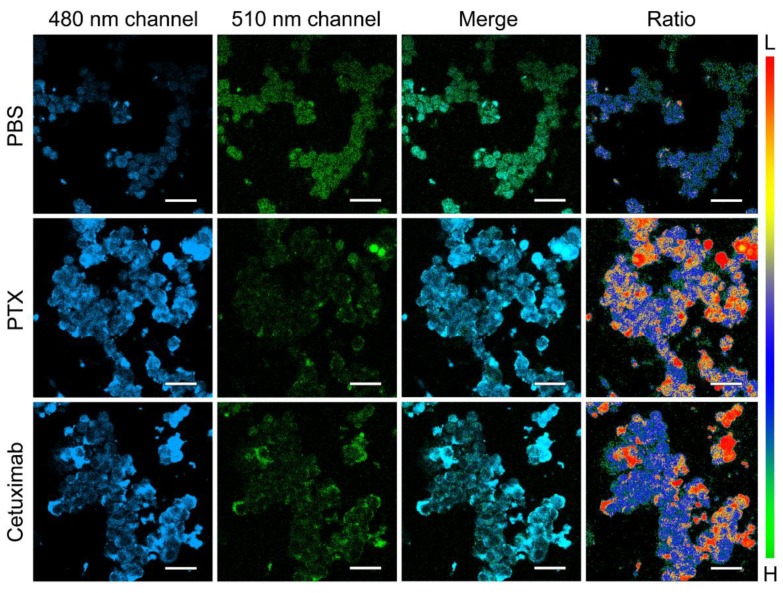
Fluorescence images (collected at 465–490 nm and 510–550 nm for two channels), merged images, and ratio images of LS180 cells incubated with TAT-ANNA upon further incubation with PBS, anti-cancer drug Paclitaxel Injection (PTX), and Cetuximab, respectively. The scale bars correspond to 10 μm.

**Figure 4 materials-12-01632-f004:**
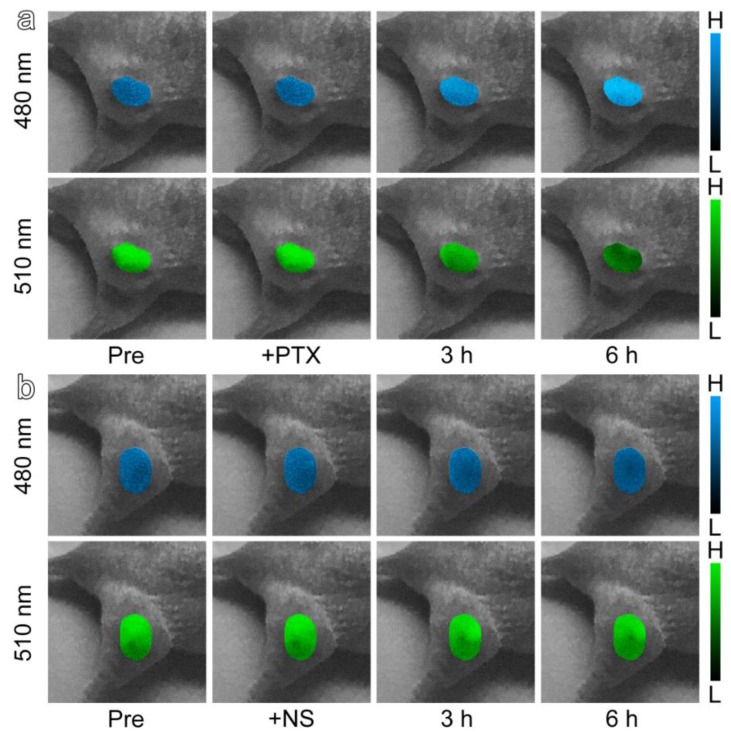
(**a**) Color-coded fluorescence images of TAT-ANNA injected into tumor-bearing mice based on the emission of blue and green channels after having been treated with (**a**) Paclitaxel Injection (PTX) or (**b**) normal saline (NS).

**Figure 5 materials-12-01632-f005:**
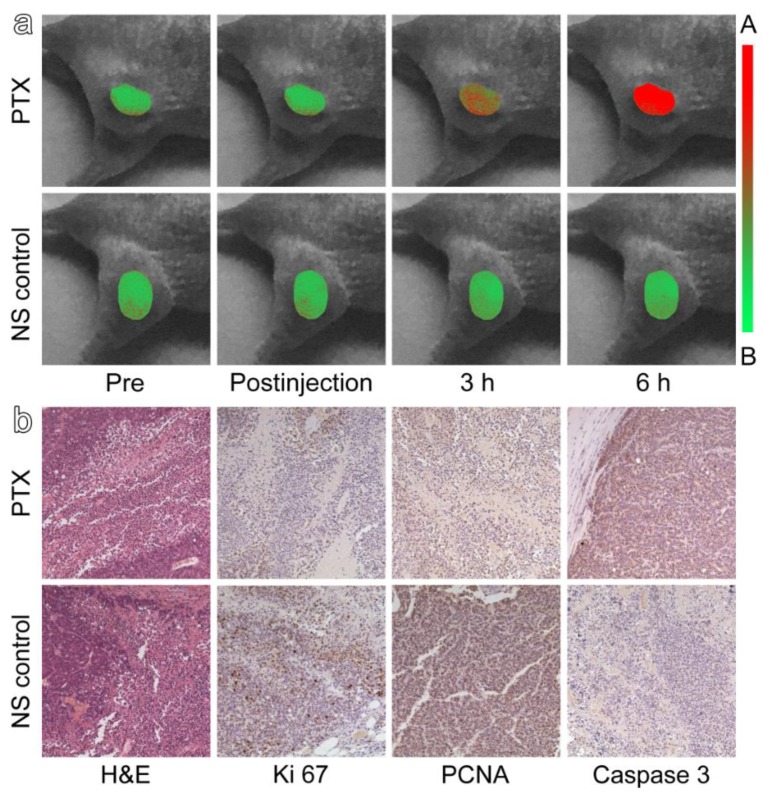
(**a**) pH mapping of the tumor regions through color-coded ratio images between the two channels (I480/I510) in Figure 4, in which red and green represent acidity and basicity, respectively. (**b**) Hematoxylin and eosin (H&E) and immunohistochemical staining images of tumor tissue slices, including Ki 67, PCNA, and Caspase-3.

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
