# Peer review of "pH-Sensitive Ratiometric Fluorescent Probe for Evaluation of Tumor Treatments"

_materials, 2019, doi:10.3390/ma12101632_

Reviewer 1 Report

The authors present an interesting set of data specifically the application of the fluorescence probe for monitoring intracellular pH both in vitro and in vivo models that would worth its publication. The manuscript is well written. However, it is not very well suited to the audience of Materials. I would advise that this could be possible in a more specialized journal, such as Biomedicines or Pharmaceutics.

Minor:

- Figure 2(b): I would suggest the author to normalize the data with the untreated sample to show the effect of the cytotoxicity.

Author Response

Replies to the Comments 

Comments:

The authors present an interesting set of data specifically the application of the fluorescence probe for monitoring intracellular pH both in vitro and in vivo models that would worth its publication. The manuscript is well written. However, it is not very well suited to the audience of Materials. I would advise that this could be possible in a more specialized journal, such as Biomedicines or Pharmaceutics.

1)    Figure 2(b): I would suggest the author to normalize the data with the untreated sample to show the effect of the cytotoxicity.

Reply: Actually, the group of cells at 0 mM concentration of TAT-ANNA is the untreated sample, the data in other groups has already normalized by this untreated one. 

Reviewer 2 Report

Paper describes preparation of pH-sensitive fluorescent probe and its further analysis in viewpoint of potential application for monitoring of anti-cancer drugs in vivo. Research issue undertaken by Authors is very interesting and undoubtedly worth investigating. In general, paper has been prepared correctly. Based on conducted research appropriate conclusions have been drawn. However, paper requires minor revisions, i.e. some issues need clarification.

Section “Introduction” should be supplemented with brief information concerning examples of currently applied ratiometric fluorescent probes.

Moreover, Authors need to emphasize the novelty of their work. There are many articles concerning pH-sensitive ratiometric probes or probes that monitor the intracellular pH. Therefore Authors should clearly indicate the novelty of presented research.

Next, in section 2.2. Authors need to describe briefly the methodology of synthesis of pH-sensitive ratiometric fluorescent dye ANNA.

In section 2.5. Authors should mention about the procedure of MTT assay- there is not any information concerning the explanation of main principles of applied assay.

Next, in section 2.10. Authors describe the methodology of treatment of tumor slices before microscopic analysis, i.e. that these were “stained with hematoxylin and eosin (H&E), Ki 67, PCNA, Caspase 3 (…)”. Authors need to explain why exactly such a treatment was applied and what are Ki 67, PCNA etc.

Manuscript needs to be re-checked grammatically and linguistically because it contains some mistakes (e.g. “were add” instead of “were added”, “were chose” instead of “were chosen” etc.).

Author Response

Replies to the Comments

Comments: Paper describes preparation of pH-sensitive fluorescent probe and its further analysis in viewpoint of potential application for monitoring of anti-cancer drugs in vivo. Research issue undertaken by Authors is very interesting and undoubtedly worth investigating. In general, paper has been prepared correctly. Based on conducted research appropriate conclusions have been drawn. However, paper requires minor revisions, i.e. some issues need clarification.

1)    Section “Introduction” should be supplemented with brief information concerning examples of currently applied ratiometric fluorescent probes.

Reply: The current applications of ratiometric fluorescent probes has already exemplified in Ref. 18-20.

2)    Moreover, Authors need to emphasize the novelty of their work. There are many articles concerning pH-sensitive ratiometric probes or probes that monitor the intracellular pH. Therefore Authors should clearly indicate the novelty of presented research.

Reply and response: We appreciate this comment. Accordingly, the revision that emphasize the novelty of our work has been added in line 242-244.

3)    Next, in section 2.2. Authors need to describe briefly the methodology of synthesis of pH-sensitive ratiometric fluorescent dye ANNA.

Reply and response: We appreciate this comment. The ANNA dye was synthesized by following our previous approach, whose details is described carefully in the new references, (Ref. 21&22 in revision).

4)    In section 2.5. Authors should mention about the procedure of MTT assay- there is not any information concerning the explanation of main principles of applied assay.

Reply: Methyl thiazolyl tetrazolium (MTT) assay is widely used as a standard method for detecting cell survival. Specifically, it is a colorimetric assay for measuring the activity of cellular succinate dehydrogenase which reduce the tetrazolium dye, MTT, to its insoluble formazan, giving a purple color, thereby reflecting the number of viable cells. MTT assays are usually used to measure cytotoxicity or cytostatic activity of potential medicinal agents and toxic materials. Additionally, the procedures of MTT assay has been provided in section 2.5 with sufficient details in the original submission.

5)    Next, in section 2.10. Authors describe the methodology of treatment of tumor slices before microscopic analysis, i.e. that these were “stained with hematoxylin and eosin (H&E), Ki 67, PCNA, Caspase 3 (…)”. Authors need to explain why exactly such a treatment was applied and what are Ki 67, PCNA etc.

Reply: Both Ki67 and PCNA are antigens related to cell proliferation, which is indispensable in cell proliferation. Proliferating cancer cells can be labeled through immunohistochemical analysis of PCNA and Ki67. Caspase 3 is one of the most convincing markers of apoptosis, over expression of which indicates that cells are suffering from apoptotic process. All these three markers are commonly used in clinical biopsy to analyze the pathological characteristics and prognosis of cancer. Therefore, these immunohistochemical analysis can offer ultimate criterions for the reliability of the TAT-ANNA probe in pharmacodynamics evaluation.

 6)    Manuscript needs to be re-checked grammatically and linguistically because it contains some mistakes (e.g. “were add” instead of “were added”, “were chose” instead of “were chosen” etc.).

Reply: We are sorry for these language mistakes. We have carefully checked the grammar mistakes throughout the manuscript. Revision has been made accordingly (one of “was” was deleted in line 85, “were added” in line 116 and “were chosen” in line 203)

Reviewer 3 Report

The manuscript is focused on the a fluorescent probe for tumor treatments imaging and its pH responsiveness as a cancer hallmark.

The topic is appropriate for the journal.

The title is adequate and correlate with the content of the article.

The abstract reports a consistent summary of the article findings, as well as the selected keywords.

The work has a clear structure.

All sections are properly written and required for a complete understanding.

Figures, tables and captions are correctly discussed.

The final summary match with the manuscript content presented and discussed.

Author Response

We appreciate these kind comments.